# Anti-Wrinkle and Dyeing Properties of Silk Fabric Finished with 2,4,6-Trichloropyrimidine

**DOI:** 10.3390/polym14163332

**Published:** 2022-08-16

**Authors:** Minhua Li, Xue Dong, Tieling Xing, Guoqiang Chen

**Affiliations:** National Engineering Laboratory for Modern Silk, College of Textile and Clothing Engineering, Soochow University, 199 Renai Road, Suzhou 215123, China

**Keywords:** 2,4,6-trichloropyrimidine, silk fabric, anti-crease finishing, reactive dyes

## Abstract

Silk, a natural protein fiber, is widely used in the textile industry and biomedical materials for its excellent properties. However, its application in some fields is seriously restricted due to its poor anti-wrinkle behavior. In this study, 2,4,6-trichloropyrimidine (TLP) was used in the production of anti-wrinkle silk fabrics. The optimum finishing conditions were as follows: 3-g/L 2,4,6-trichloropyrimidine, 6-g/L NaHCO_3_, 8-g/L Na_2_SO_4_, finishing temperature of 65 °C, and finishing time of 40 min. The crease recovery angle of the finished fabric is 16–20% higher than the unfinished fabric, and the finishing process has a small effect on the whiteness of silk while achieving some degree of washing resistance. The morphology and chemical structures of the finished silk fabric were characterized by scanning electron microscopy and Fourier transform infrared spectroscopy. The K/S value of the finished silk fabric dyed with reactive dyes increased compared with the silk fabric only dyed, indicating that the dyeability of the finished fabric was improved. This technology provides a new method for fabricating silk color crease-resistant fabrics.

## 1. Introduction

Silk fabric, a green natural protein product, is applied to home textiles, clothing, and other fields due to its outstanding properties, such as smooth feel, soft luster, good moisture absorption, comfortable wearing, and eco-friendliness. At the same time, due to its good mechanical properties, biocompatibility, degradability, and drug permeability, it has been increasingly used in the field of biomedical materials. However, silk fabric is easy to wrinkle, has poor light stability, and other shortcomings, which affect its feel, appearance, and wearing performance, and limit its application [1,2]. Therefore, how to improve the anti-wrinkle performance of silk fabric on the premise of retaining its original excellent characteristics is of great importance to increase the added value of silk products and expand the application field of silk products.

Silk fibroin has low crystallinity, several voids exist in its interior, and there are many salt and hydrogen bonds in its amorphous region, lacking certain chemical cross-linking. When water and other solvent molecules interact with silk fabrics, they irreversibly destroy hydrogen and salt bonds between protein polymers and deform the fibrin amorphous region, causing the silk fabric to wrinkle [1,3]. Chemical cross-linking with anti-wrinkle agents is one of the most commonly used methods to improve the wrinkle resistance of silk [4,5].

Several studies have been conducted on cross-linked anti-wrinkle finishing agents, mainly including resins, polycarboxylic acids, waterborne polyurethanes, reactive silicones, and epoxy compounds. Traditional anti-wrinkle finishing agents, such as two-dimensional resin can significantly improve the anti-wrinkle effect of silk fabrics, but there is a formaldehyde release problem during its application, which endangers laborer’s health [6,7,8]. A silk fabric finished using polycarboxylic acid-based finishing agents, such as 1,2,3,4-butane tetracarboxylic acid (BTCA) achieved satisfactory resilience, whiteness, and washability [2,9]. Nevertheless, silk anti-crease finishing with BTCA has the disadvantages of significant strength loss and high cost. Other common formaldehyde-free silk anti-wrinkle agents, such as epoxides [10,11], siloxanes [12], glyoxal [13,14,15], glutaraldehyde [16], and natural or synthetic polymers [17,18,19] have been studied, exhibiting some crease-resistant effects. However, they have various disadvantages. Specifically, it is very necessary to study the formaldehyde-free and environmentally friendly anti-wrinkle agent suitable for silk fabric.

2,4,6-Trichloropyridine (TLP) is an heterocyclic compound containing nitrogen, which has been successfully applied in various fields due to its pharmacological and strong biological activities [20]. Halogen-pyrimidine reactive dyes are widely used in the printing and dyeing industry. For example, 2,4,6-trichloropyrimidine is often used as an intermediate in medicine and reactive dyes. However, to the best of our knowledge, few studies have been conducted on their use in textile finishing.

A new anti-wrinkle agent for silk fabric was studied in this paper. The finishing agent can be used to prepare wrinkle-resistant fabrics under mild conditions, and they will be formaldehyde-free during deposition or wearing.

## 2. Materials and Methods

### 2.1. Materials and Reagents

Silk crepe de chine (52 g/m^2^) was supplied by Suzhou Taihu Snow Silk Co., Ltd. (Suzhou, China). The silk fabrics were used after soap boiling at 60 °C for 30 min. TLP was supplied by Shanghai Adama Reagent Co., Ltd. (Shanghai, China). Sorbitan monolaurate (Span 20), polyoxyethylene-dehydrated sorbitol monooleate (Tween 80), sodium sulphate (Na_2_SO_4_), and sodium bicarbonate (NaHCO_2_) were supplied by Shanghai Aladdin Biochemical Technology Co., Ltd. (Shanghai, China). Foam-free powder was purchased from Shenggong Biological Engineering Co., Ltd. (Shanghai, China). Reactive dyes (blue DS) were provided by Shanghai Anoky Group Co., Ltd. (Shanghai, China). All reagents were analytically pure and used without any further purification.

### 2.2. Preparation and Characterization of Dispersed Emulsions

TLP can be emulsified using the two following methods to prepare stable TLP dispersion emulsion:

Emulsification method 1: Evenly mix TLP with Span 20 (20% owf) and Tween 80 (30% owf). After obtaining the mixture, gradually add water and stir evenly. Then, emulsify for 30 min using the mechanical dispersion method to finally obtain a 3-g/L solution.

Emulsification method 2: Evenly mix TLP with Span 20 (20% owf) and Tween 80 (30% owf). After obtaining the mixture, gradually add water and stir evenly. Then, emulsify for 40 min using the ultrasonic dispersion method to finally obtain a 3-g/L solution.

### 2.3. Preparation of TLP-Finished Silk

According to the technological process in Figure 1, the silk fabric is immersed in dispersing emulsions of different configurations in an oscillating dyeing machine. Different finishing agent concentrations, different alkaline concentrations, different finishing times, and different finishing temperatures were studied under the condition of the best application method. The liquor ratio of the fabric to TLP dispersion emulsion was set to 1:50. Then, the fabric was dipped in the dispersion emulsion for 45 min. Finally, the finished silk fabric was subsequently washed with deionized water for 15 min, followed by sample drying and room temperature preservation.

### 2.4. Dyeing of Silk

The control and finished silk fabrics were dyed with reactive dyes using exhaustion to observe the dyeing rate and efficiency for the exploration of the anti-wrinkle property of dyed fabric. The concentration of the dyeing solution was 2% owf, and the ratio of silk weight to the dye solution was maintained at 1:50. The fabrics were dyed at 60 °C for 45 min after a temperature rise from room temperature to 60 °C at a rate of 1–2 °C/min. NaHCO_3_ (7 g/L) and Na_2_SO_4_ (10 g/L) were added gradually in four instalments at a 10-min interval in the dyeing process. After dyeing, the dyed fabrics were withdrawn and washed with distilled water until no color change occurred. For this experiment, the reactive dye blue DS was used as the dyeing agent, and different dyeing and finishing processes were applied: Method 1: Dyeing and finishing; Method 2: Dyeing, finishing, and bathing; Method 3: Finishing and dyeing.

### 2.5. Characterization and Measurements

#### 2.5.1. Finishing Agent Emulsion Particle Size Test

The finishing agent emulsion was diluted to 1% solution with deionized water. The particle size and distribution coefficient of the emulsion were determined at 25 °C by Malvern Nano ZS90 particle size/potential analyzer (Malvern & Company, London, UK).

#### 2.5.2. Anti-Wrinkle Properties of Silk Fabrics

According to the AATCC66-2003 Test Method, the crease recovery tester (YGB 541E, standard equipment) was used to measure the crease recovery angle of silk samples in warp and weft directions. The longitude and latitude of each sample were measured five times, then the average value was calculated and added. Tests were performed under standard laboratory conditions (20 ± 1 °C, 65 ± 5% R.H) [21].

#### 2.5.3. The Surface Morphology and Structure of the Finished Fabric

The surface morphology of silk samples was observed with a desktop scanning electron microscope (Hitachi TM 3030, Tokyo, Japan) and the test voltage was maintained at 15 kV. Before observation, all samples were coated with conductive sputtering gold. The Fourier transform infrared spectrum (FTIR) of the sample was recorded using an FTIR instrument (Nicolet 5700, Thermo Fisher Scientific Inc., New York, NY, USA) at a KBr ball in the wave number range of 500–4000 cm^−1^. Scanning was conducted 16 times.

#### 2.5.4. Durability to Washing

According to the AATCC61-2006 Standard Method, the washing durability of silk samples was tested on the Wash Tec-P Fastness Tester (Cockroach International, Kent, UK). Each sample was immersed in a solution containing 2 g/L commercial detergent at 40 °C for 5 min at a ratio of commercial detergent to sample of 50:1. After each wash, the fabric was squeezed gently, and then rinsed with fresh deionized water [22]. For each new washing cycle, new detergent is added to ensure the existence of surfactants.

#### 2.5.5. Mechanical Properties

The tensile strength was evaluated using the Instron 3365 Universal Testing Machine (Instron, MI, USA) according to ISO 13934-1-2013 with a specimen of 300 mm × 50 mm. The test was performed five times for each sample [23] under standard laboratory conditions (20 ± 1 °C, 65 ± 5% R.H).

#### 2.5.6. Whiteness Index

The whiteness index (WI) was determined according to the AATCC11-2005 Testing Method. The average of four readings was noted from different positions of the sample using a Datacolor 650^®^ Bench-Top Spectrophotometer [21]. Tests were performed under standard laboratory conditions (20 ± 1 °C, 65 ± 5% RH).

#### 2.5.7. Color Measurements

Color characteristic values (L*, a*, b*), whiteness, and color depth (K/S) value of the dyed fabrics were examined using a Hunter Lab Ultra Scan PRO reflectance spectrophotometer with a small area view and D65 primary source. The absorbance of the dye solution at different wavelengths was measured using a TU-1900 UV–Vis spectrophotometer to determine the maximum absorption wavelength of the dye solution. The absorbances A0 and an of the dye solution before and after dyeing, respectively, were measured at the maximum absorption wavelength, and the dye uptake rate of the dye solution was calculated using Equation (1).
(1)$$\mathrm{Dyeing}\;\mathrm{rate}=1-\frac{A_{n}}{A_{0}}\times 100\%$$

## 3. Results and Discussion

### 3.1. Effect of Particle Size of Finishing Agent Emulsion on Crease Recovery Angle of Finished Silk Fabrics

Emulsified finishing liquids with different particle sizes were obtained using the two emulsifying methods mentioned in Section 2.2. The anti-crease finishing of real silk fabric was performed according to the anti-crease finishing process mentioned in Section 2.3. The finishing temperature was 95 °C, and the finishing time was 40 min. The effect of particle size of finishing agent emulsion on dry and wet crease recovery of the finished silk fabric was examined. The results are shown in Table 1.

As shown in Table 1, compared with Method 1 (mechanical dispersion method), the finishing agent emulsion obtained using Emulsification Method 2 (ultrasonic dispersion Don’t keep method) has a small particle size, small particle size dispersion coefficient, and narrow particle size distribution. After finishing the silk fabric using Emulsification Method 2 (ultrasonic dispersion method), the recovery angle of the dry and wet crease is larger. The results show that the finishing agent emulsion with a small particle size can be prepared through ultrasonic dispersion, which can improve the anti-crease finishing effect of silk fabrics. This is due to the fact that the finishing agent emulsion has a small particle size and narrow particle size distribution. The finishing agent emulsion can easily diffuse into the fabric, even between silk fibers, and has more cross-linking reactions with silk fibers, improving the anti-crease performance of the finished silk fabric. Therefore, Emulsification Method 2 was used to emulsify TLP to prepare the finishing agent emulsion.

### 3.2. Effects of Reaction Conditions on Anti-Wrinkle of TLP-Finished Silk Fabrics

To prepare silk with good crease resistance, the effects of different TLP dispersants, various alkaline agents and their concentrations, Na_2_SO_4_ concentration, finishing product time, and temperature conditions on the anti-crease performance of silk will be investigated to determine the best finishing product process. Figure 2 shows the anti-wrinkle effect of silk under different reaction conditions. As shown in Figure 2a, when TLP dispersion is in the range of 1–5 g/L, the crease resistance of the finished silk fabric increases gradually with the increasing concentration. However, when the TLP concentration exceeds 3 g/L, the crease recovery angle almost does not increase or even decreases slightly. To reduce cost, the concentration of the finishing agent is 3 g/L, which is optimal. An alkaline agent provides an alkaline environment and is one of the most important factors in the final finishing process. Under acidic and neutral conditions, only –OH reaction occurs on silk, whereas, in an alkaline environment, both –NH and –OH of silk are covalently bonded with TLP. However, peptide bonds in protein fiber macromolecules are easily hydrolyzed under alkaline conditions, affecting the strength of the fiber. Therefore, to preserve fiber quality, protein fiber should be prepared under a mild pH condition. Therefore, NaHCO_3_ was used as the base agent. As can be seen in Figure 2b, with the increase in NaHCO_3_ concentration, the pH value of finishing solution increases, and the anti-crease performance of silk increases first. When NaHCO_3_ concentration is 6 g/L, the pH value of the finishing solution is 9, and the wet crease recovery angle of silk begins to decrease. Therefore, as an alkaline agent, the finishing solution is weakly alkaline, and 6 g/L NaHCO_3_ is the most suitable. As shown in Figure 2c, under the condition of TLP concentration of 3 g/L and NaHCO_3_ concentration of 6 g/L, the temperature required for the reaction was changed, the finishing effect of anti-crease of real silk is the best under the condition of medium temperature. TLP-based finishing of silk can be performed at medium temperature (65 °C). In terms of the finishing time, under the conditions of TLP concentration of 3 g/L, NaHCO_3_ concentration of 6 g/L, reaction temperature of 65 °C, as shown in Figure 2d, after finishing for 40 min, the anti-crease performance of real silk decreased slightly with the increasing finishing time. Therefore, 40 min is optimal.

### 3.3. Surface Morphology and Chemical Structure

We observed the morphology of the fabric surface by scanning electron microscopy. The surface morphology of the control and finished silk fabrics is shown in Figure 3. The surface of the unfinished fabric is smooth, whereas the surface of the finished fabric is covered with several fine attachments and is partially rough. This surface roughness can restrict the relative movement of silk fabric macromolecules by mechanical friction and is useful to the wrinkle resistance of silk.

The chemical structure of the finished silk fabric was analyzed by FTIR and X-ray photoelectron spectroscopy. Infrared (IR) spectra of the finished and control silk fabrics are shown in Figure 4. The peaks at 1617 and 1512 cm^−1^ represent the amide I and II structures of silk fiber, respectively [24]. The wet and dry crease recovery angles of the finished fabric are higher than the unfinished silk fabric. It is reasonable to infer that the finished fabric is cross-linked with TLP. The most likely cross-links should be formed by chlorine atoms in TLP reacting with hydroxyl or amino groups in silk. This conjecture can be supported by the enhancement of the peak near 1050 cm^−1^ in the IR spectrum, which is attributed to the stretching vibration of the C–N or C–O–C bond [25].

### 3.4. Washing Durability and Whiteness

To evaluate the water washing resistance of the finished silk fabric, several pieces of TLP treated silk were washed 10 and 20 times. The durability of the washed finished fabric is evaluated in terms of the wrinkle recovery angle. It can be seen from the results in Table 2 that the crease recovery angle of the silk fabric is reduced by about 10° after washing for 10 times, and by about 20° after washing for 20 times. Compared with the unfinished fabric, the silk fabric still has better crease resistance. Additionally, the measured standard deviation (SD) is between 3.9 and 6.3, indicating that the results are universal.

In terms of fabric strength, with the increase in finishing agent TLP concentration, the cross-linking between TLP and silk fabric increases, in order that the macromolecules of silk fabric are not easy to slip, and the breaking strength decreases. It can be seen from Table 3, when TLP concentration increased to 4 g/L, the fracture strength of silk fabrics decreased by 3.7% compared with unfinished fabrics, and the SD was 0.8, which does not affect the subsequent use performance of silk fabric. As can be seen in Table 4, five tests were carried out on fabrics finished with different concentrations of TLP. Through the one-way analysis of variance (ANOVA), the F-value is greater than 1, indicating that the concentration of finishing agent is the main factor affecting the change of fabric strength when other conditions are fixed.

As for the whiteness of fabric, it can be seen in Table 3 and Table 4 that the whiteness of unfinished fabric is 69.9, and its SD is 1.8. The whiteness value of the fabric after TLP treatment decreased by 1.5, and its SD was 1.7. This indicates that the color of the treated silk fabric is even. At the same time, due to the high finishing temperature, the whiteness of the finished fabric decreases slightly, which does not affect the appearance of the fabric. Four whiteness tests were carried out on fabrics finished with different concentrations of TLP. Through the one-way ANOVA, it can be seen in Table 5 that the F-value is less than 1, indicating that the difference between test objects does not have a great impact on the results, only the random error influence, thus the sample is uniform.

### 3.5. Dyeing Properties

For this experiment, the reactive dye blue DS was used as the dyeing agent. Table 6 shows the K/S value, color parameters, washing fastness, and dyeing rate of the treated and control silk. The L*, A *, B *, C*, and H values of the treated silk fabric are comparable to those of the control silk fabric, indicating that the treated silk fabric has little influence on the dyeing of silk. Where, L* represents light and shade (black and white), a* represents red and green, b* represents yellow and blue, c* represents chroma (the degree of saturation or purity of color), and H represents tone Angle. The K/S value of silk fabric dyed and finished is lower than the silk fabric only dyed without finishing. This is due to the fact that a deeper color can still be obtained after increasing the dye concentration. However, the K/S value and dyeing rate of the fabric after finishing were significantly increased, indicating that the affinity of the reactive dye was improved after finishing [24]. The washing fastness of the treated and untreated fabrics slightly changed. Therefore, TLP finishing can increase the affinity of the fabric to the dye without affecting its washing fastness.

The crease resistance of dyed and finished fabrics was tested. As shown in Figure 5, the crease resistance of both dyed and finished fabrics (Method 1) and washed and finished fabrics (Method 2) and finished and dyed fabrics (Method 3) increased by more than 15% compared with only dyed fabrics. Among them, finishing and dyeing (Method 3) can improve the crease resistance of the fabric by 27%. It shows that dyeing has little effect on the anti-crease property of the finished fabric.

In addition, three kinds of commercially available reactive dyes (blue DS) were used in calculating the dyeing rate of different dyes by different dyeing and finishing processes at the same dye concentration, according to the calculation method of dyeing rate described in 2.5.7. As shown in Table 7, the dyeing and finishing one-bath method (Method 2) reduced the dyeing rate of the three types of dyes compared with the dyeing and finishing first (Method 1), since there was a competitive relationship between dyes and finishing agents and the combination of fabrics in the one-bath dyeing and finishing process. Compared with dyeing before finishing (Method 3), the dyeing rate of dyeing and finishing (Method 1) increased, indicating that the affinity of finished fabrics to reactive dyes increased. Specifically, the affinity of the finished fabric to reactive dyes increases, and washing fastness has little effect on the finished fabric.

## 4. Conclusions

In this study, silk fabrics were treated and finished with TLP compounds, and an energy-saving anti-wrinkle finishing process was obtained by impregnation. The optimal finishing process conditions are as follows: TLP concentration of 3 g/L, NaHCO_3_ concentration of 6 g/L, Na_2_SO_4_ concentration of 8.0 g/L, finishing temperature of 65 °C, and finishing time of 40 min. In addition, TLP can work with dyes to reduce the dyeing and finishing process time. The results showed that the finishing process did not lead to clear damage to the physical and mechanical properties of the silk fabric, and the fold recovery angle of the finished silk fabric was increased by 16–20% compared with the unfinished fabric. Furthermore, the finished silk fabric shows good hand feeling, whiteness, and dyeing properties. TLP could be a promising formaldehyde-free anti-wrinkle finishing agent for silk.

## Figures and Tables

**Figure 1 polymers-14-03332-f001:**
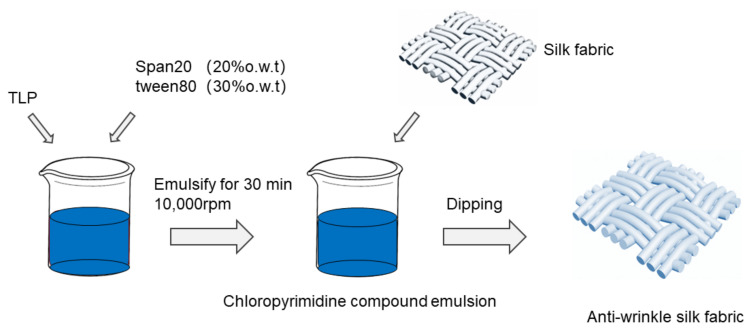
Schematic of the preparation process of anti-wrinkle silk fabric.

**Figure 2 polymers-14-03332-f002:**
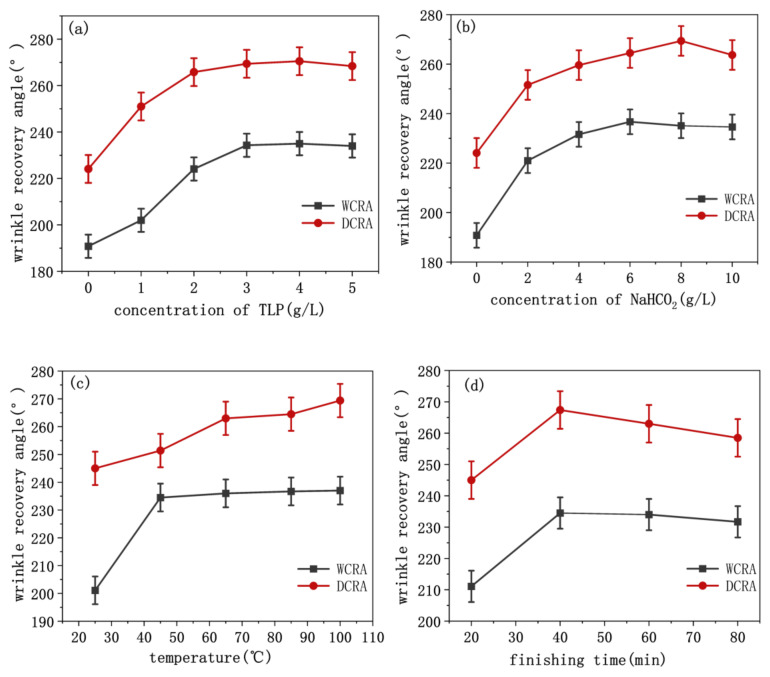
Effects of reaction condition on anti-wrinkle of TLP-finished silk fabric. (**a**) Concentration of TLP dispersions; (**b**) concentration of NaHCO_2_; (**c**) finishing temperature; (**d**) finishing time.

**Figure 3 polymers-14-03332-f003:**
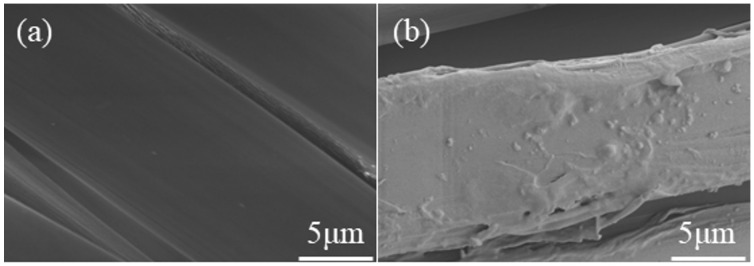
Scanning electron microscopy images of (**a**) control and (**b**) TLP-finished silk fabrics.

**Figure 4 polymers-14-03332-f004:**
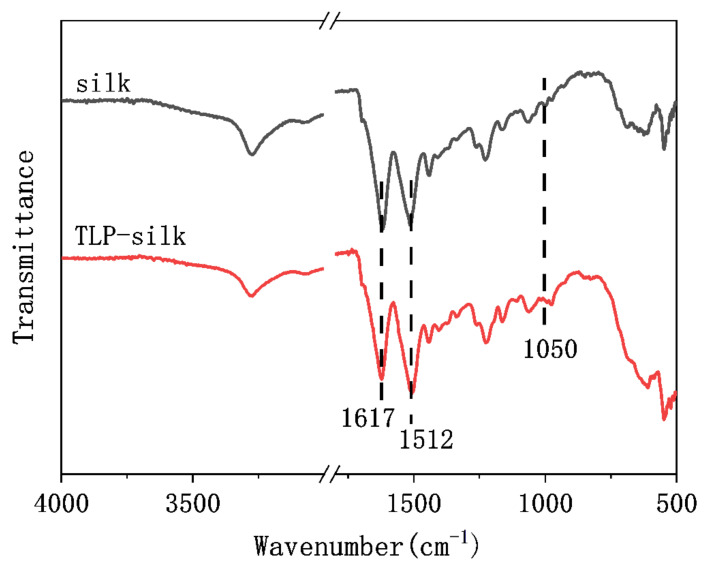
FTIR spectra of control and TLP-finished silk fabrics.

**Figure 5 polymers-14-03332-f005:**
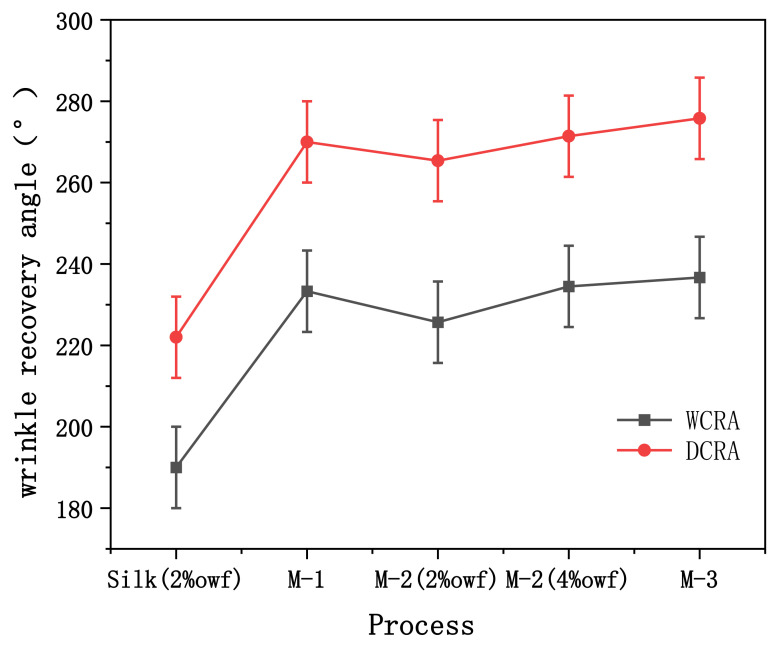
Wrinkle resistance of dyed fabrics.

**Table 1 polymers-14-03332-t001:** Effect of particle size of finishing agent emulsion on crease recovery angle of finishing silk fabric.

The Sample	Emulsification Method	Mean Particle Size (nm)	Dispersion Coefficient	DCRA (°)	WCRA (°)
1	Method 1	1450.2	0.435	257	221
2	Method 2	287.9	0.183	284	234

**Table 2 polymers-14-03332-t002:** Washability of finished fabric.

		Before Washing	SD	10 Washings	SD	20 Washings	SD
Silk	WCRA (°)	198.5 ± 4.5	4.6	185.3 ± 4.1	3.9	172.8 ± 5.0	4.9
	DCRA (°)	223.4 ± 6.1	5.9	208.5 ± 5.7	5.6	195.2 ± 5.6	5.5
TLP-silk	WCRA (°)	247.2 ± 5.6	5.2	238.6 ± 6.2	6.3	220.3 ± 6.7	6.3
	DCRA (°)	268.1 ± 4.8	4.7	256.1 ± 4.4	4.2	241.6 ± 4.9	4.9

**Table 3 polymers-14-03332-t003:** Fracture strength retention rate and whiteness of finished and control silk.

	Fracture Strength Retention Rate (%)	Whiteness WI CIE [D65/10]
Silk	-	69.9 ± 2.3
TLP-silk (2 g/L)	98.7 ± 1.2	68.5 ± 1.9
TLP-silk (3 g/L)	97.6 ± 0.7	68.7 ± 1.7
TLP-silk (4 g/L)	96.3 ± 1.1	68.4 ± 2.0

**Table 4 polymers-14-03332-t004:** Strong retention analysis of variance.

	The First Test	The Second Test	The Third Test	The Fourth Test	The Fifth Test	Average Value
1	99.5	98.2	97.9	99.0	98.9	98.7
2	98.6	96.4	97.1	98.3	97.6	97.6
3	95.2	96.9	97.4	95.9	96.1	96.3
The total average	97.53					
MSA	21.65					
MSE	1.45					
F(MSA/MSE)	14.93 > 1					

**Table 5 polymers-14-03332-t005:** Analysis of variance for whiteness.

	The First Test	The Second Test	The Third Test	The Fourth Test	Average Value
1	68.1	69.7	70.4	65.8	68.5
2	68.4	65.9	69.9	70.6	68.7
3	70.4	68.8	65.7	68.7	68.4
The total average	68.53				
MSA	0.28				
MSE	1.57				
F(MSA/MSE)	0.18 < 1				

**Table 6 polymers-14-03332-t006:** K/S value, color parameters, and washing fastness of finished and control silk fabrics.

Process	K/S	Colour Parameter					Washing Fastness
		L*	a*	b*	C*	H	
Silk (2%owf)	11.38	32.53	−7.65	−16.64	18.32	245.31	4
Method 1	11.45	32.8	−7.52	−16.89	18.49	246.01	4
Method 2 (2%owf)	7.61	39.87	−10.63	−17.18	20.2	238.25	3–4
Method 2 (4%owf)	12.36	30.7	−6.98	−16.34	17.45	247.27	3–4
Method 3	14.08	28.88	−5.97	−15.8	16.89	249.3	4–5

**Table 7 polymers-14-03332-t007:** Dye uptake of different types of dyes in different dyeing and finishing processes.

Type of Dye	Process	Dyeing Rate (%)
Monochloro triazine type	Method 1	83.6
vinyl sulphone type		85.8
double reactive type		89.6
Monochloro triazine type	Method 2	58.7
vinyl sulphone type		75.1
double reactive type		83.0
Monochloro triazine type	Method 3	85.1
vinyl sulphone type		87.9
double reactive type		91.6

## Data Availability

Not applicable.

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
