# Peer review of "Anti-Wrinkle and Dyeing Properties of Silk Fabric Finished with 2,4,6-Trichloropyrimidine"

_polymers, 2022, doi:10.3390/polym14163332_

Round 1

Reviewer 1 Report

Abstract

- silk is a fibre, not a fabric and can be made in many different constructions

- correct grammar

Text

- silk is a small % of world fibre consumption so not really 'highly popular'

- pilling - not a common problem with silk as it depends more on yarn structure and fibre length

- 2.4 - define type of water

2.5.1 - provide supplier of equipment

2.5.2 - provide ambient conditions

- define statistical analysis (seems not to have been completed/ reported)

3.2 wording - Therefore....

3.4 there are sections in several parts which are Methods, not Results so need to be re-located

Several sections refer to energy saving/environmental issues - re-word or provide references and values

Tables 2, 3 show mean results but there is no analysis for significance of differences (Friedman for non-parametric data, ANOVA for interval data) - needs to be undertaken and reported

References

- several incomplete without page numbers - amend; use of lower case and capitals irregular - amend; usefful to include DOI for all publications - check and add

-

Author Response

Response to reviewer 1 Comments

Thank you very much for your comments, which are all valuable and helpful for revising and improving our paper, as well as the important guiding significance to our research. We have studied comments carefully, and the revised portion is marked in red in the paper, some of the content is provided in the supporting information. The main corrections in the paper and the responds to the reviewer’s comments are as follows:

Point1:silk is a fibre, not a fabric and can be made in many different constructions

Response:Thank you for this valuable feedback. The silk fabric used in this study is electric spinning and its crease resistance needs to be improved.

Point2: silk is a small % of world fibre consumption so not really 'highly popular'

- pilling - not a common problem with silk as it depends more on yarn structure and fibre length.

Response: Thank you for this suggestion and we have revised the adjective and used the word strictly.

Point3: - 2.4 - define type of water

2.5.1 - provide supplier of equipment

2.5.2 - provide ambient conditions

Response: we apologize for the confusion caused. We have supplemented the content to make the explanation clearer. The revised content was as follows:

-2.4- cold distilled water

-2.5.1- Malvern Nano ZS90 particle size/potential analyzer.

-2.5.2- The samples were placed under standard laboratory conditions (20 ℃ and 65% R. H) for 24 h before testing.

Point4:  -3.3-there are sections in several parts which are Methods, not Results so need to be re-located.

Response: We extremely grateful to you for pointing out this problem. We subsequently supplemented the conclusions drawn from table 2: The results in Table 2 show that, the crease recovery Angle of the finished silk fabric decreased by 20° after 20 times of washing, and it still had better crease resistance compared with the unfinished fabric.

Point5: Several sections refer to energy saving/environmental issues - re-word or provide references and values.

Response: Thank you for your careful review and evaluation. The reaction condition is medium temperature, compared with the high temperature required by the traditional anti-wrinkle agent (polyurethane, citric acid, etc.) finishing, relatively more energy saving. As for environmental friendliness, there is no strong evidence to support it. We will correct the wording.

Point6: Tables 2, 3 show mean results but there is no analysis for significance of differences.

Response: We thank Reviewer 1 for these insightful comments. As for table 2, we supplemented the comparison of wrinkle resistance of TLP treated fabric after 20 times of washing with that of untreated fabric; With regard to table 3, we added the reason why the breaking strength of silk fabrics decreased compared with that of untreated fabrics with the increase of finishing agent concentration. All samples were measured several times to obtain the average value. Suitable for most silk fabrics with different structures.

Point7: several incomplete without page numbers - amend; use of lower case and capitals irregular - amend; useful to include DOI for all publications.

Response: We agree with this suggestion. We will complete it

Reviewer 2 Report

This paper gives valuable information on the formaldehydes free anti-wrinkle finishing and dyeing for silk.  Thus, I recommend this paper to be published in Polymers with minor revision.

I recommend describing the relationship between wrinkle recovery angle and pH instead of the concentration of NaHCO2, since the purpose of adding NaHCO2 in the finishing bath is to adjust pH of the solution.

Because optimal temperature is closely related with pH of the bath, please specify the pH of the bath in Figure 2(c ).

Various dyeing method described in the Results and Discussion section on page 8 should be moved to Materials and method section 2.4.

You should provide wrinkle recover angles as well as dyeing rate according to different dyeing methods in Table 5 and explained about competition of dyes and TLP, since the main purpose of this study is to improve durable press properties of silk with TLP.

Author Response

Response to reviewer 1 Comments

Thank you very much for your comments, which are all valuable and helpful for revising and improving our paper, as well as the important guiding significance to our research. We have studied comments carefully, and the revised portion is marked in red in the paper, some of the content is provided in the supporting information. The main corrections in the paper and the responds to the reviewer’s comments are as follows:

Point1: describing the relationship between wrinkle recovery angle and pH instead of the concentration of NaHCO3

Response: Thank you for this valuable feedback. We will mark the pH values corresponding to different concentrations of NaHCO3, and analyze the relationship between pH values and wrinkle resistance of the fabric. In this study, the optimal concentration of NaHCO3 is 6g/L and the pH is 9. Therefore, pH 9 is one of the best reaction conditions.

Point2: Various dyeing method described in the Results and Discussion section on page 8 should be moved to Materials and method section 2.4.

Response: Thank you for pointing this out. We move the various staining methods described in Section 3.5 to Section 2.4.

Point3: You should provide wrinkle recover angles as well as dyeing rate according to different dyeing methods in Table 5 and explained about competition of dyes and TLP.

Response: we apologize for the confusion caused. We have added the test data on the wrinkle resistance of silk fabrics after dyeing and finishing in different order. The crease resistance of dyed and finished fabrics was tested. As shown in Figure 5, the crease resistance of both dyed and finished fabrics (Method 1) and washed and finished fabrics (Method 2) and finished and dyed fabrics (Method 3) increased by more than 15% compared with only dyed fabrics. Among them, finishing and dyeing (method 3) can improve the crease resistance of the fabric by 27%.

 It shows that dyeing has little effect on the anti-crease property of the finished fabric.

Round 2

Reviewer 1 Report

Introduction - silk cannot be popular given such a small % of world fibre produced - re-word.

Methods - 2.5.2 - reference needed; state whether or not the tests were conducted under these standard conditions.

Results - the analysis of significance of differences in results observed is required (simply looking at figures/images is insufficient, needs analysis of variance or other suitable tests).

Author Response

Response to Reviewer 1 Comments

Thank you for your constructive comments for our paper. we believe that we have been able to address your comments and suggestions, and that our paper has been substantially improved as a result. Below we indicate how we responded to each of your comments.

Point1:Introduction - silk cannot be popular given such a small % of world fibre produced.

Response:We agree with this suggestion. We changed the word " is popular in home textiles, clothing and other fields " to " is applied to home textiles, clothing and other fields."

Point2:- 2.5.2 - reference needed; state whether or not the tests were conducted under these standard conditions.

Response:We apologize for the confusion caused. We added the literature that the test was performed under this condition.

[21] Jiangfei Lou, Xuerong Fan, Qiang Wang, Ping Wang, Jiugang Yuan, Yuanyuan Yu, Oxysucrose polyaldehyde: A new hydrophilic crosslinking reagent for anti-crease finishing of cotton fabrics, Carbohydrate Research, Volume 486,2019,107783.

Point3: the analysis of significance of differences in results observed is required (simply looking at figures/images is insufficient, needs analysis of variance or other suitable tests).

Response:Thank you for this valuable feedback. We have added the variance of the whiteness of the finished fabric to further illustrate that the finishing agent has little effect on the whiteness of the fabric and the whiteness of the fabric is uniform.

The whiteness variance of the finished fabric fluctuates around 1, indicating that the finished fabric is more uniform in color, which further proves that the finishing agent has little influence on the whiteness of the fabric.

Round 3

Reviewer 1 Report

Thank you for making some of the changes suggested.

2.3 - various??? needs further detail to demonstrate optimum

2.5.5 - need to specify whether or not all tests were conducted under standard conditions

Methods/Results - still some methods in the Results section - revise (e.g. see p8)

There is still no analysis of the results to determine whether these are significant or simply chance. This is required.

Author Response

Response to Reviewer 1 Comments

Thank you for your constructive comments for our paper. we believe that we have been able to address your comments and suggestions, and that our paper has been substantially improved as a result. Below we indicate how we responded to each of your comments.

Point1:2.3 - various??? needs further detail to demonstrate optimum

Response:We apologize for the confusion caused. We modified it to: "Studied in different finishing agent concentration, different alkaline concentration, different finishing time and different finishing temperature under the condition of the best application method."

Point2:2.5.5 - need to specify whether or not all tests were conducted under standard conditions

Response:Thank you for this valuable feedback. We added test criteria: The tensile strength was evaluated using Instron 3365 machine according to ISO 13,934–1–2013 with a specimen of 300 mm by 50 mm and the test was performed five times for each sample.

Point3: Methods/Results - still some methods in the Results section - revise (e.g. see p8)

Response: We agree with this suggestion. We have removed the method in 3.5 Staining properties.

Point4: There is still no analysis of the results to determine whether these are significant or simply chance. This is required.

Response: I'm very sorry and I don't understand about you put forward about what part of the results of the analysis without detailed analysis. All the experimental results have been verified many times and are not accidental.

Round 4

Reviewer 1 Report

There remain several changes required:

• Methods

- state whether all physical/mechanical testing was conducted under standard conditions. You state pre-conditioned for 24 hours (also need to provide reference for this as it seems not to be ISO) but omit to state the conditions for the actual testing (2.5.2, 2.5.5, 2.5.6)

- there is nothing on statistical analytical methods, and no statistical analysis results (as per my previous comment). It is not  sufficient to look at the graphs and make concluding comments without providing the stats test results.

• Results

- see note above re statistical analysis required

Author Response

Response to Reviewer 1 Comments

Thank you for your constructive comments for our paper. we believe that we have been able to address your comments and suggestions, and that our paper has been substantially improved as a result. Below we indicate how we responded to each of your comments.

Point1:- state whether all physical/mechanical testing was conducted under standard conditions. You state pre-conditioned for 24 hours (also need to provide reference for this as it seems not to be ISO) but omit to state the conditions for the actual testing (2.5.2, 2.5.5, 2.5.6)

Response:We agree with this suggestion. We recalibrated the criteria of the test:

2.5.2, According to AATCC test Method 66-2003, the crease recovery tester (YGB 541E, standard equipment) was used to measure the crease recovery Angle of silk samples in warp and weft directions. The longitude and latitude of each sample were measured 5 times, the average value was calculated and added up. Tests were performed under standard laboratory conditions (20℃±1℃, 65%±5% R.h).

We added the literature that the test was performed under this condition.

[21] Jiangfei Lou, Xuerong Fan, Qiang Wang, Ping Wang, Jiugang Yuan, Yuanyuan Yu, Oxysucrose polyaldehyde: A new hydrophilic crosslinking reagent for anti-crease finishing of cotton fabrics, Carbohydrate Research, Volume 486,2019,107783.

2.5.5, The tensile strength was evaluated using Instron 3365 machine according to ISO 13,934–1–2013 with a specimen of 300 mm by 50 mm and the test was performed five times for each sample. Tests were performed under standard laboratory conditions (20℃±1℃, 65%±5% R. h). We added the literature that the test was performed under this condition.

[23]Xian-Wei Cheng, Shuang Dong, Hai-Jun Yang, Li-Ping Zhao, Jin-Ping Guan,The development of phosphorus-doped hybrid silica sol coating for silk with durable flame retardancy,Polymer Degradation and Stability,Volume 201,2022,109974.

2.5.6,The whiteness index (WI) was determined according to AATCC Testing Method 11–2005. The average of four readings was noted from different positions of the sample using a Datacolor 650®Bench-Top Spectrophotometer

We added the literature that the test was performed under this condition.

[21] Jiangfei Lou, Xuerong Fan, Qiang Wang, Ping Wang, Jiugang Yuan, Yuanyuan Yu, Oxysucrose polyaldehyde: A new hydrophilic crosslinking reagent for anti-crease finishing of cotton fabrics, Carbohydrate Research, Volume 486,2019,107783.

Point2:- there is nothing on statistical analytical methods, and no statistical analysis results (as per my previous comment). It is not sufficient to look at the graphs and make concluding comments without providing the stats test results.

Response:We apologize for the confusion caused. We added error values to the test results in Table 2 and Table 3 and improved the data analysis. AATCC test method 66-2003 was used to test the crease recovery Angle after washing in Table 2. Longitude and latitude were measured 5 times for each sample. The breaking strength in Table 3 was assessed according to ISO 13934-1-2013 with 5 tests per sample; Whiteness index (WI) in Table 3 as measured by AATCC test Method 11-2005 recorded 4 readings from different locations of the sample.

Table 2. Washability of finished fabric.

Before washing

10 washings

20 washings

Silk

WCRA (°)

198.5±4.5

185.3±4.1

172.8±5.0

DCRA (°)

223.4±6.1

208.5±5.7

195.2±5.6

TLP-silk

WCRA (°)

247.2±5.6

238.6±6.2

220.3±6.7

DCRA(°)

268.1±4.8

256.1±4.4

241.6±4.9

Table 3. Fracture strength retention rate and whiteness of finished and control silk.

Fracture strength retention rate(%)

whiteness WI CIE [D65/10]

Silk

-

69.86±2.3

TLP-silk(2 g/L)

98.7±1.2

69.46±1.9

TLP-silk(3 g/L)

97.6±0.7

68.74±1.7

TLP-silk(4 g/L)

96.3±1.1

68.45±2.0

This manuscript is a resubmission of an earlier submission. The following is a list of the peer review reports and author responses from that submission.